# Feature Matching Synchronized Reasoning from Energy-Based Memory Network for Intelligent Data Management in Cloud Computing Data Center

**JeongYon Shim**

Division of GS, Computer Science, KNU College, Kangnam University, Yongin Si 449-702, Korea;
mariashim@kangnam.ac.kr; Tel.: +82-31-280-3736

**Abstract:** A cloud data center for software-as-a-service (SaaS) was built for the purpose of stably managing these server computers in one place in order to provide an uninterrupted service, not only for a stable power supply and security but also for the efficient data management. To manage such a data center efficiently, it is important to build a cloud database with structured storage above all else. In recent decades, many studies have focused on designing cloud data centers and most of the research has focused on communication traffic, routing, topological issues and communication technology. However, in order to build an efficient cloud database that can support user demand, the most sophisticated intelligent system, based on AI technology and considering user convenience, should be designed. From this viewpoint, adopting human brain functions, Energy-Based Memory Network was designed for a knowledge-based frame of an intelligent system. And its event-related synchronized data extraction mechanism was proposed. In particular, a Thinking Thread extraction phase was implemented for the reasoning process using qualia matching and a deep extraction method in a cloud database. The purpose of this approach is to design and implement an intelligent cloud database that has an efficient structure and mechanism for supporting user demand and providing accurate, prompt services. In experiments, the working phase of the functions was simulated with data and analyzed. As a result, it was confirmed that the proposed system works well and intelligently for the design purpose.

**Keywords:** cloud database; energy-based knowledge network; qualia matching; deep extraction

## 1. Introduction

A cloud data center is a building or facility that provides server computers and network lines and they have grown with the spread of the Internet. As tens of thousands of server computers were needed to store and display vast amounts of information such as internet searches, and shopping, games and education websites, data centers were built for the purpose of stably managing these server computers in one place. An uninterrupted providing service, a stable power supply and security are important. Most data centers are installed in multistory high-rise buildings. A cage is installed for each user group on each floor, several racks are installed in the cage and a switch is placed in each rack to operate several server computers [1].

Software-as-a-service (SaaS) is simply 'Software provided through the cloud'. To explain in detail, it can be interpreted as 'a service provided through the Internet by software installed in the public cloud without a separate installation or conversation process'. In the past, to use software in a physical hardware infrastructure such as a PC or server was absolutely necessary. Installing SW on hardware and using it has been common sense for the past 30 years. However, the development of the Internet has shattered this common sense. The disappearance of installed software and the replacement of SaaS is an irresistible trend in the software industry. Therefore, many software developers who used to develop installed software are converting their software to SaaS. Google

initially developed all of its service as SaaS through Gmail ID and Microsoft is also rapidly converting its installed software to SaaS, such as the SaaS conversion of MS office including document creation, email management, MS Dynamics and an ERP/CRM solution. Artificial intelligence, which is emerging as a hot topic in the IT industry, can play an important role in API-type SaaS. The most important thing is to provide the best service to meet users' demands. SaaS (Software-as-a-Service) should be able to provide the necessary information and services immediately by providing accurate information. It is necessary to build the intelligent cloud data center operated by a well-structured database and intelligent mechanism [2].

To manage such a data center for SaaS efficiently, above all, it is important to build a cloud database with structured storage. A large amount of data must be managed systematically and, for this purpose, it is necessary to build an intelligent system based on AI technology that considers user convenience [3].

For recent decades, many studies on topology for network, communication technology, load balancing, routing strategies, storage, database and big data mechanism, etc., have been conducted to attempt to implement a cloud data center [4]. Especially in cloud computing, the porting of AI technology [5] is starting to be performed for intelligent operation [6].

In order to build an efficient cloud database, first, it is necessary to structure the data storage based on intelligence. The data structure should be easily accessible to store the memory and extract the related data selectively from the viewpoint of the user. Second, an intelligent mechanism should be embedded to manage this. Because Cloud Database deals with a large amount of data with high complexity, we should design an appropriate intelligent mechanism sophisticatedly and efficiently.

We can find a clue in human brain structure and functions to find a solution. Since the human brain is a very smart and efficient organ that has been evolved for millions of years in the complex and ever-changing real world, adopting the brain functions is a good approach to provide a decisive way for solving the problem.

Therefore, the purpose of this approach is to design a more efficient, intelligent system for user-oriented cloud data. That is, we propose the structure and the working intelligent mechanism of a cloud database that acts like brain memory. In Section 2, we study the human brain structure and the functions of synapses that form memories and examine the working principle of ERD/ERS among brain mechanisms, drawing out the possibility of its application. In Section 3, Energy-Based Memory Network, which is structured as a cloud database, is designed for a knowledge-based frame of an intelligent system. In this system, a Thinking Thread extraction phase is processed using qualia matching and a deep extraction method. The working phase of the functions is simulated with data and described in Section 4.

## 2. Related Works: Cloud Computing Database and the Functions of Brain Memory

### 2.1. Cloud Computing Database Strategy in the Data Center

Cloud computing is the on-demand delivery of computing services, such as databases, storage, servers, networking, software, analytics and intelligence. These services are delivered over the Internet or 'cloud' for nearly instantaneous access to critical business data and resources. Instead of owning their own data center or computing infrastructure, businesses can rent access to storage, applications and other services from a cloud service data center.

There are three main types of cloud computing: Software-as-a-Service (SaaS), Infrastructure-as-a-Service (IssS) and Platform-as-a-Service (PaaS).

SaaS is one of the most popular forms of cloud computing. This method of software delivery makes data to be easily accessed from any device with an internet connection and web environment. It is a complete solution that users can buy from a cloud service provider on a pay-as-you-go basis. Cloud computing gives a great option and benefit for users who want to save money, enhance their performance, promote scalability, improve collaboration

and keep business [7]. For satisfying users' demands, there needs some preconditions and considerations: For the expected future cost compared to the cost of owning their own infrastructure, their storage capacity needs, security expectations and the anticipated level of support. should be provided [8].

A data center used for cloud computing is a physical organization and physical facility that organizations use to house their data and applications [9]. A data center's design is based on a network of computing and storage resources to enable the delivery of shared applications and data [10].

In recent years, as interest in cloud computing has increased, a lot of research has been conducted; most of this research has been on the network configuration and topology constituting the data center, its scalability, the smooth flow of data streams [11], and routing mechanisms [12], etc. [13]. As one topological model, Figure 1 demonstrates the topological construction of Fat-free and Jellyfish networks [14]. Jellyfish is designed to have rigid structure of a high-capacity network interconnection, which yields itself naturally to incremental expansion with flexible switching [15]. The configuration of Maglev with load balancing, shown in Figure 2, is also one example of the related works [16].

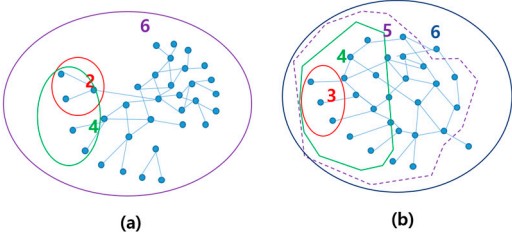

**Figure 1.** Random Graph of (**a**) Fat-free (**b**) Jellyfish.

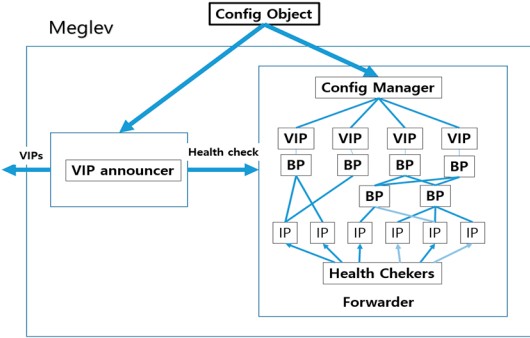

**Figure 2.** The Configuration of Maglev with Load Balancing.

A signal-oriented data stream management system that can sense phenomena with both signal processing [17] and asynchronous event stream processing was proposed for memory manager and scheduler optimization tuned for processing signal segments [18].

However, while the amount of data streams to be processed in data centers is explosively increasing, user demands are becoming more personalized. To solve this problem, intelligent techniques are desperately needed. A well-defined network topology with a complicated and flexible data structure should be implemented and an efficient data-storing strategy based on this infrastructure is necessary. In order to maximize the service for users' needs, a personalized high-level data extraction mechanism should be supported that accurately analyzes user requirements by designing well-defined queries and individual characteristics.

From this point of view, the intelligent design of this data center is very important to provide users with high quality and expected services. Data center cloud services are a natural fit for processing big data streams because they allow data mining algorithms to run at a scale for handling uncertain data volume, variety, velocity and personalization.

### 2.2. Human Brain and Its Function: Memory Formation in the Neural Network

The brain is made up of numerous nerve cells called neurons that communicate with each other through synapses as shown in Figure 3 [19]. Synapses are the basic units of information that connect neurons to other neurons and mainly serve as a connection switching for signal transmission in the neural networks [20]. For the brain to operate properly, the synapses that make up the neural circuits must work well [21]. The major adhesion protein PTPσ is found in specific neural circuits in the hippocampus and is known to mediate memory and regulate function by helping localize and stabilize NMDA receptors [22]. Synapses can continuously change their structure and function according to their activities and play an important role in memory formation [23]. During information processing, the corresponding synapses are activated flexibly in the neural networks [24].

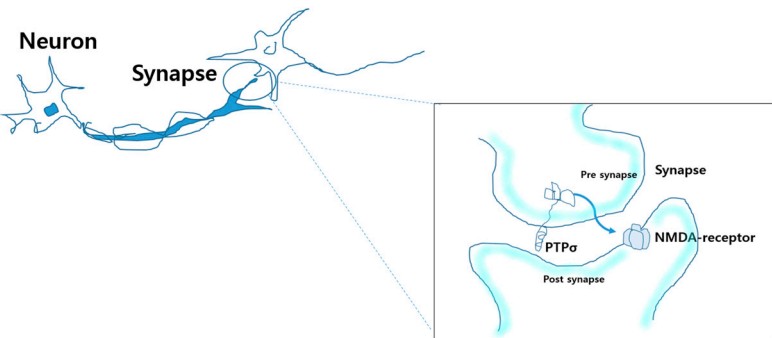

**Figure 3.** Neuron and Synapse of Neural Network in Human Brain.

### 2.3. Event-Related Synchronization

EEG (electroencephalography) interface technology is called Brain–Computer Interface, a technology that enables an interface with a computer using brain waves. ERD (event-related desynchronization) and ERS (event-related synchronization) are generally accepted as valid EEG variables for EEG commands [25]. Events not only in the generation of an event-related potential (ERP) but also in a change in the ongoing EEG in the form of ERD or ERS. ERD refer to a phenomenon in which the alpha region rapidly decreases before human operation. ERS refers to the increase in the appearance of the Beta region when the operation starts. While the former is phase-locked, the latter is not phase-locked to the event. ERD and ERS can be observed at nearly the same time as the localization of cortical areas is involved in task-relevant processing [26].

Conflicting EEG changes based on the operation timepoint suggest the possibility of extracting EEG commands. That is, this represents the possibility that EEG commands can also affect on brain waves. Figure 4 shows the ERD/ERS signals in time, which are displayed simultaneously.

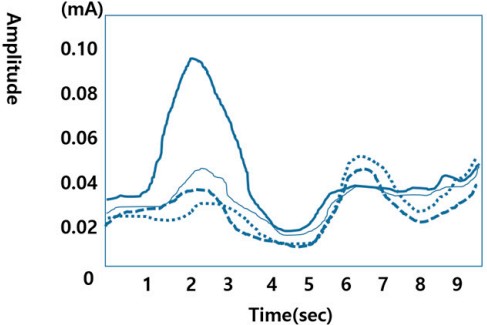

**Figure 4.** ERS/ERD Signal.

### 3. Energy-Based Memory Network System Design for Cloud Database and ERS Data Extraction Mechanism

In this section, an energy-based memory network system for a cloud database is designed for an intelligent platform adopting the functions of memory formation and event-related synchronization and including a topological concept of network design. This system has an energy-based memory network as a basic frame of stored memory and an event-related synchronized data extraction mechanism that can extract the related data with ERS signal from an energy-based memory network.

### 3.1. Energy-Based Memory Network System Design for Cloud Database

#### 3.1.1. System Overview

As shown in Figure 5, the proposed energy-based memory network consists of a learning scheme for memory formation, a data extraction phase and a reasoning scheme. In this work, we will not describe the learning scheme but focus on the data extraction phase and reasoning scheme from a predefined energy-based memory network, assuming that memory has been already formed by the learning process. The energy-based memory network as a system knowledge base is designed to consist of a memory capsule and connection.

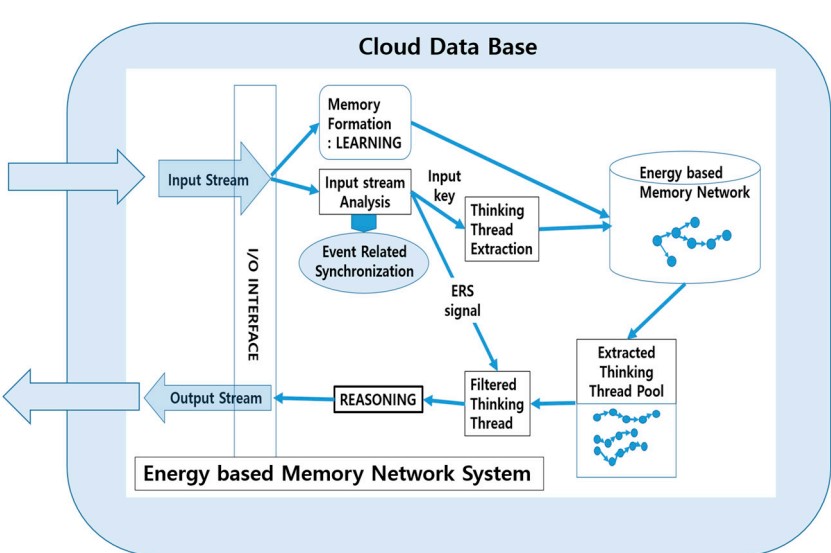

**Figure 5.** The Structure of an Energy-based Memory Network System as a Cloud Database.

#### 3.1.2. System Workflow

The workflow of an energy-based memory network system starts from the input stream that consists of a user's requests and the related data in the cloud computing environment after an energy-based memory network was set and prepared for the extracting and reasoning phase. Incoming input streams are analyzed through the 'Input Stream Analysis' module working with the 'Event-Related Synchronization' module. As a result of analysis, an 'Input keyword' and 'ERS signal' are produced. The produced 'Input keyword' flows into the 'Thinking Thread Extraction' module for retrieving the related Thinking Threads from the 'Energy-based Memory Network'. During this process the data extraction mechanism works and the related Thinking Threads are retrieved. The extracted Thinking Threads are separated into a single path with no branches and are dispatched to and stored in the 'Extracted Thinking Thread Pool'. This pool is temporarily created in every extracting phase. The 'Filtered Thinking Thread' module filters the extracted Thinking Threads in the previous step taking the ERS signal from 'Input Stream Analysis' module and produces filtered Thinking Threads. In the final step, the 'REASONING' module works and provides output streams of results which the user wants from I/O INTERFACE.

### 3.2. The Basic Frame of an Energy-Based Memory Network

### 3.2.1. The Structure of an Energy-Based Memory Network

An energy-based memory network is defined as a basic frame of knowledge base structure for a reasoning phase in this proposed system. The memory network is designed to consist of a memory capsule and connection link as shown in Figure 6 [27]. The memory capsule as a basic memory cell has important information such as ID (identification), E (Energy value) and A (attributes).

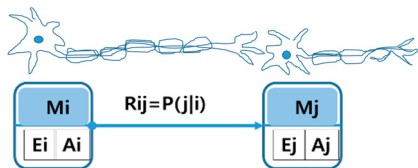

**Figure 6.** Memory Capsule and the connection link $R_{ij}$ between $M_i$ and $M_j$.

The attributes of the memory capsule, A, representing its property have $P_i$ (Probability of Self) and $Q_i$ (qualia) term. This information is used for extracting the data very importantly. Each of the memory capsules is connected to each other by connecting wires which are named as connection links. Connection link, $R_{ij}$, represents the connection strength between the memory capsules $M_i$ and $M_j$ and has a value calculated by Equation (1) [28].

$$R_{ij} = P(i|j) \tag{1}$$

Each memory capsule has an energy, $E_i$, which represents the energy state measured by positive degree and excited state in the energy space as in Figure 7. The energy value is expressed in the $(x, y)$ coordinates from of energy space such as Equation (2).

$$E_i = (x_i, y_i) \tag{2}$$

where $x_i$ is a value representing positive degree(P) if $x_i$ is greater than zero, neutral state(U) if $x_i$ is equal to zero and negative degree(N) if $x_i$ is less than zero. In addition, where $y_i$ is a value representing excited(C) state if $y_i$ is greater than zero, neutral state(U) if $x_i$ is equal to zero and inhibitory state(I) if $y_i$ is less than zero.

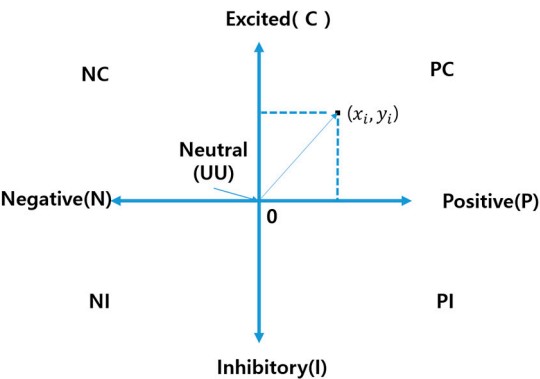

**Figure 7.** Energy State.

Energy value, $E_i^v$ in the energy state $(x_i, y_i)$ is calculated by Equation (3).

$$E_i^v = d\sqrt{(x_i{}^2 + y_i{}^2)} \tag{3}$$

$$d = \begin{cases} 1.0 & if\ P, C, U \\ -1.0 & else\ N, I \end{cases}$$

Additionally, energy state is represented as PU (positive neutral), PC (positive excited), PI (positive inhibitory), NU (Negative neutral), UU (Neutral), UX (neutral excited), UI (neutral inhibitory), PI (positive inhibitory) and NI (negative inhibitory). These values are used for the total energy calculation of the extracted Thinking Thread.

### 3.2.2. Qualia Concept for Feature Matching

In philosophy and certain models of psychology, qualia are defined as individual instances of subjective, conscious experience [29]. There are many different definitions depending on perspectives, but in this work we define 'qualia' as characteristics of things, i.e., 'of what sort' or 'of what kind' in a specific instance such as 'what it is like to' [30]. Therefore, we designed the qualia term as a component of the attribute in memory capsule (which is a basic unit of knowledge as defined by this system) and used the qualia concept for feature matching during the data extracting process. We classified the state of qualia to five features: V (growing and expanding like a tree), F (active like fire), D (decreasing and shrinking), G (hard and unchanging as gold) and W (smooth and calm as water). The number of terms and their characteristics can be decided flexibly depending on the system design. For the feature matching process, all the data used in this system should be predefined to have qualia terms.

Figure 8 shows qualia matching degrees between each term which are designed for feature matching in this proposed system.

|   | V | F | D | G | W |
|---|---|---|---|---|---|
| **V** | 1.0 | 0.5 | 0.0 | -0.5 | 0.5 |
| **F** | 0.5 | 1.0 | 0.0 | 0.5 | -0.5 |
| **D** | 0.0 | 0.0 | 1.0 | 0.0 | 0.0 |
| **G** | -0.5 | 0.5 | 0.0 | 1.0 | 0.5 |
| **W** | 0.5 | -0.5 | 0.0 | 0.5 | 1.0 |

**Figure 8.** Qualia Matching Rule.

Qualia matching is used for Thinking Thread generation during the event-related synchronized data extracting process. In this table, the qualia matching degree, q, between V and F is marked as 0.5. It means that the qualia matching degree between V and F, 0.5, not only represents a reactive value but also controls the degree of activation during thread extraction phase on the energy-based memory network.

### 3.2.3. Memory Storage Cycle and Management

Data storing processes and maintenance in this system are performed by a strategy driven by the cycle specified in Figure 9. The memory management mechanism consists of 'Creation', 'Operating', 'Wake/Sleep', 'Repairing', 'Removing' and 'Incarnation' steps. The system monitors the memory states and performs the appropriate work according to the monitoring result for keeping an efficient memory. A more detailed description about memory storage management is explained in the paper [31].

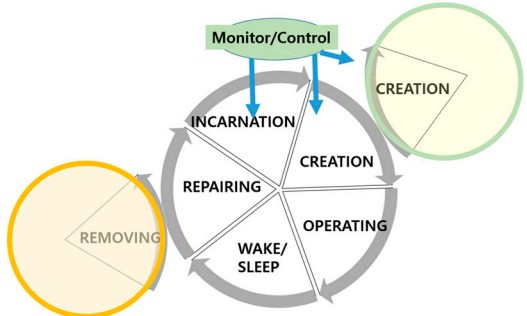

**Figure 9.** Memory Storage Management Cycle.

### 3.3. Event-Related Synchronized Data Extraction Mechanism

3.3.1. Input Stream and Input Stream Analysis

As shown in Figures 5 and 10, the input stream coming through the I/O interface propagates to the data analysis module in which incoming data are preprocessed for the appropriate process in this system. The incoming input stream, which consists of input data, data type and data strength. When the users enter data, they can select one using a drop-down button. With regards to data type, users can select one of V, F, D, G, W and Unknown, and for data strength, they can choose one of High, Medium, Low and Unknown. These entered data are polished using a connected 'INDataBase' so that they can be handled well by the system. INDataBase contains 'KeyList', 'Qtype' and 'E' areas. The 'Data Analysis Module' checks for synonyms and the data form using the KeyList produces the 'InputKey'. Using the Qtype base, qualia type is checked and an ERS signal is produced. The data analysis module changes the incoming data from Data Strength to a 'DataEnergy' value in the $(x, y)$ coordinate form.

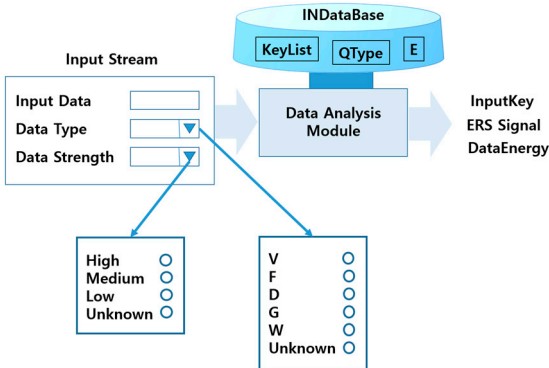

**Figure 10.** Input Stream Analysis.

3.3.2. Thinking Thread Extraction

In this section, the Thinking Thread extraction phase from an energy-based network is described. Because an energy-based network has a form of network, the InputKey value passed from the previous step propagates to the connected nodes starting from the activated node. During the process of traversing the network, the related Thinking Threads are extracted. In this phase, the single path without a branch as a Thinking Thread is retrieved. The Thinking Thread extraction module extracts all the related Thinking Threads from the energy-based memory network and passes to the extracted Thinking Thread pool. Because the extracted Thinking Thread pool is a temporary storage, it temporarily keeps only the extracted Thinking Threads produced by the current cycle of the Thinking Thread extraction mechanism.

Thinking Thread extraction requires an efficient data structure for easy access and network traversal. As a structure to realize this, the Memory Network List has been designed and its form is as follows:

Memory Network List:

$$\left[M_i, E_x^i, E_y^i, P_i, Q_i, R_{ij}, M_j\right] = \left[M_i, E_I, A_i, R_{ij}, M_j\right]$$

where $E_i = [E_x^i, E_y^i]$ and $A_i = [P_i, Q_i]$.

This notation can be interpreted as the memory capsule, $M_i$, which has an energy value $(E_x^i, E_y^i)$, self probability, $P_i$, and qualia type, $Q_i$ and is connected to the memory capsule, $M_j$ with the strength of $R_{ij}$. Figure 11 shows an example of Energy-Based Memory Network and Table 1 describes its converted Memory Network List

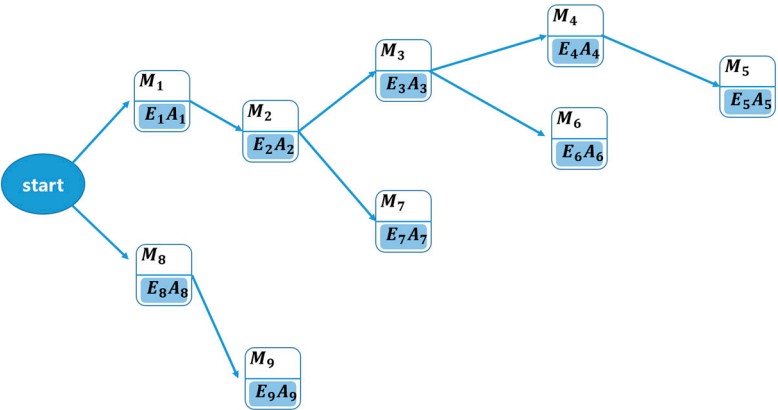

**Figure 11.** An example of Energy-Based Memory Network.

**Table 1.** Memory Network List.

| Node i | Ex | Ey | Pi | Qi | Rij | Node j |
|--------|-----|-----|-----|-----|------|--------|
| start | 1.0 | 1.0 | 1.0 | 1.0 | 1.0 | $M_1$ |
| start | 1.0 | 1.0 | 1.0 | 1.0 | 1.0 | $M_8$ |
| $M_1$ | $E_x^1$ | $E_y^1$ | $P_1$ | $Q_1$ | $R_{12}$ | $M_2$ |
| $M_2$ | $E_x^2$ | $E_y^2$ | $P_2$ | $Q_2$ | $R_{23}$ | $M_3$ |
| $M_2$ | $E_x^2$ | $E_y^2$ | $P_2$ | $Q_2$ | $R_{27}$ | $M_7$ |
| $M_3$ | $E_x^3$ | $E_y^3$ | $P_3$ | $Q_3$ | $R_{34}$ | $M_4$ |
| $M_3$ | $E_x^3$ | $E_y^3$ | $P_3$ | $Q_3$ | $R_{36}$ | $M_6$ |
| $M_4$ | $E_x^4$ | $E_y^4$ | $P_4$ | $Q_4$ | $R_{45}$ | $M_5$ |
| $M_5$ | $E_x^5$ | $E_y^5$ | $P_5$ | $Q_5$ | 0.0 | NILL |
| $M_6$ | $E_x^6$ | $E_y^6$ | $P_6$ | $Q_6$ | 0.0 | NILL |
| $M_7$ | $E_x^7$ | $E_y^7$ | $P_7$ | $Q_7$ | 0.0 | NILL |
| $M_8$ | $E_x^8$ | $E_y^8$ | $P_8$ | $Q_8$ | $R_{89}$ | $M_9$ |
| $M_9$ | $E_x^i$ | $E_y^i$ | $P_i$ | $Q_i$ | 0.0 | NILL |

The extracted Thinking Threads by the Thinking Thread extraction mechanism are as follows:

T1:    $M_1$ $M_2$ $M_3$ $M_4$ $M_5$
T2:    $M_1$ $M_2$ $M_3$ $M_6$
T3:    $M_1$ $M_2$ $M_7$
T4:    $M_8$ $M_9$

The total energy of the Thinking Thread, $T_i$, $E_t(T_i)$, is calculated by Equation (4)

$$E_t(T_i) = \sum_{k=1}^{n} d\sqrt{\left(E_x^{i\,2} + E_y^{i\,2}\right)} \tag{4}$$

$$d = \begin{cases} 1.0 & if\ P, C, U \\ -1.0 & elif\ N, I \end{cases}$$

The value of the total energy of the Thinking Thread represents the certainty strength of memorization or belief. During the reasoning process, in case multiple Thinking Threads are retrieved, this value can be usefully used for the selection of thread. The algorithm of Thinking Thread Extraction is as Algorithm 1.

| **Algorithm 1: Thinking Thread Extraction** |
|---|
| 1:       enter the InputKey |
| 2:       search M_Id matching with InputKey |
| 3:       if Found: |
| 4:          EXTRACTION(InputKey, Mnet) |
| 5:       else: |
| 6:          exit |
| 7:       stop |
| // EXTRACTION function |
| 8:       EXTRACTION(InputKey, MNet): |
| 9:       for index range(1,1,m): |
| 10:          M_ID_flag=False |
| 11:      while(True): |
| 12:         if(EOF): |
| 13:            exit |
| 14:         else: |
| 15:           if InputKey != M_Id: |
| 16:              exit |
| 17:           else: |
| 18:              while(not EOF): |
| 19:                if M_Id_flag=False |
| 20:                   put M_Cap to QUEUE |
| 21:                   M_Id_flag=True |
| 22:                   M_Cap=M_Cap.next |

3.3.3. Qualia Matching ERS Thinking Thread Extraction and Reasoning

As described in Section 3.3.2, the related Thinking Threads are extracted through the Thinking Thread extraction module. The extracted Thinking Threads are stored in the extracted Thinking Thread pool for more sophisticated processing and reasoning. The filtered Thinking Thread module performs filtering of event-related synchronized threads activated by ERS signal. For each extracted Thinking Thread, this module calculates the matching value between the qualia term, $Q_i$, of each memory capsule and the ERS signal of the input stream. The ERS signal would be one of V,F,D,G and W. The qualia matching degree between $Q_i$ and the ERS signal can be obtained by referring to the table in Figure 9. During this process, only memory capsules that have a qualia matching degree greater than 0.0 can be activated. If there exist inactivated memory capsules inside the Thinking Thread, the strength of connection link, $R_{ij}$, should be changed. This value can be adjusted by Equation (5).

$$R_{ij} = \min\left(R_{ik},\ R_{ks,..}R_{uj}\right) \tag{5}$$

In the extraction step, the energy value of the memory capsule activated by qualia matching, $E_i^v$, is changed to a new value, $E_i^{v'}$, that reflects the reactivity by Equation (6)

$$E_i^{v'} = 0.1 \times q_i \times E_{ii}^v + E_i^v \tag{6}$$

where $q_i$ is the degree of qualia matching between the input stream and the current memory capsule. Depending on the qualia term and energy value of the input stream, $E_{ii}^v$, the response degree of the memory capsule varies. This means that when extracting a Thinking Thread from an energy-based memory network, the qualia term and energy value of the input stream are reflected and extracted.

The ERS Thinking Thread extraction mechanism of this system controls the degree of extraction with three options: qualia matching using ERS signals, deep extraction by depth δ and energy strength θ. The reasoning process is performed based on the Thinking Threads extracted in this way.

The ERS Thinking Thread extraction and reasoning mechanism is described in Algorithm 2 in detail.

| Algorithm 2: ERS Thinking Thread Extraction and Reasoning |
| --- |

| | |
| --- | --- |
| 1: | enter the InputKey, ERS signal, Extraction depth & Energy Strength θ. |
| 2: | while(True): |
| 3: | if(EOF): |
| 4: | exit |
| 5: | else: |
| 6: | copy Thread to Tlist |
| 7: | if option == QM: |
| 8: | QualiaMatching |
| 9: | if option == DE |
| 10: | DeepExtraction |
| 11: | if option == ES: |
| 12: | print(OutStream(QM,DE)) |
| 13: | else: |
| 14: | print(OutStream(QM,DE)) |
| 15: | else: |
| 16: | print(OutStream(QM,Nill)) |
| 17: | else: |
| 18: | print(OutStream(Nill,Nill) |
| // Qualia Matching | |
| 19: | QualiaMatching: |
| 20: | enter ERS signal |
| 21: | while(True): |
| 22: | if(EOF): |
| 23: | exit |
| 24: | else: |
| 25: | QualiaMatching |
| 26: | Change Energy Value of Memory Capsule, $E_i^{v\prime} = q_i \times E_i^v$ |
| // DeepExtraction | |
| 27: | DeepExtraction: |
| 28: | enter extraction depth, δ |
| 29: | while(True): |
| 30: | if(True): |
| 31: | exit |
| 32: | else: |
| 33: | if $R_{ij} \geq \delta$ : |
| 34: | $R_{ij} = R_{ij}$ |
| 35: | else: |
| 36: | $R_{ij} = 0.0$ |

## 4. Experiments

In our experiments, the functions of event-related synchronized data extraction are simulated with a virtual energy-based memory network. Focusing on qualia matching and Thinking Thread extraction, we tested the proposed mechanism with data and given values as shown in Table 2. Additionally, we investigated the reactive results of qualia matching state and energy value in the point of qualia matching and extracting depth. Figure 12 shows a master energy-based network testbed that consists of 13 memory capsules. In this figure, we marked the energy, attribute and connection link term as an abbreviated form. Figure 13 describes qualia matching with an incoming ERS signal (V, F, D, G, W) of input stream and its reactive change of network. In the case of (a) and (b), they represent qualia matching with incoming V type ERS signal and its change, respectively. We expressed the qualia matching degree by coloring. The blue colored node with 'START' mark means a starting point. The dark blue painted nodes, lightly blue painted nodes, unpainted nodes and orange painted nodes represent its qualia matching degree of greater than zero, zero and less than zero, respectively. These terms decide the extracting feature during the Thinking Thread extraction process. In the figure, dark or light blue painted nodes represent activated states, unpainted nodes represent inactivated states and orange painted

nodes mean inhibitory states. Therefore, only blue painted nodes and connection links are filtered from a master energy-based memory network. When the node is inactive, the value of the connection link is adjusted by Equation (5). In the case of orange colored nodes of inhibitory state, the nodes connected to it are excluded during the Thinking Thread extraction process.

**Table 2.** Memory Capsule list: Memory ID, Energy value, $P_i$ (Probability of Self), $Q_i$ (Qualia).

| $M_i$ | $E_x^i$ | $E_y^i$ | $P_i$ | $Q_i$ |
|-------|---------|---------|-------|-------|
| $M_1$ | 1.0 | 0.7 | 1.0 | V |
| $M_2$ | 0.9 | 0.6 | 1.0 | F |
| $M_3$ | 0.8 | 0.8 | 1.0 | D |
| $M_4$ | 0.5 | 0.6 | 1.0 | G |
| $M_5$ | 0.9 | 0.3 | 1.0 | V |
| $M_6$ | 0.5 | 0.7 | 1.0 | W |
| $M_7$ | 0.9 | 0.8 | 1.0 | F |
| $M_8$ | 0.4 | 0.6 | 1.0 | G |
| $M_9$ | 0.8 | 0.2 | 1.0 | F |
| $M_{10}$ | 0.4 | −0.3 | 1.0 | W |
| $M_{11}$ | 0.5 | 0.3 | 1.0 | V |
| $M_{12}$ | 0.7 | 0.7 | 1.0 | G |
| $M_{13}$ | 0.6 | 0.5 | 1.0 | D |

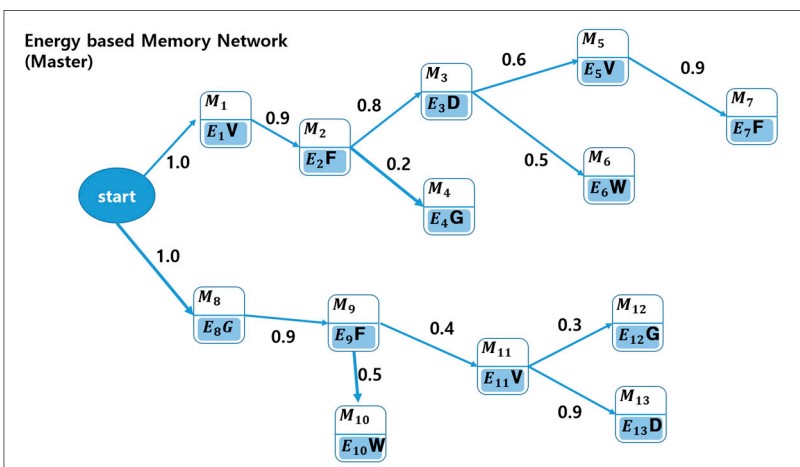

**Figure 12.** Master Energy-based Memory Network.

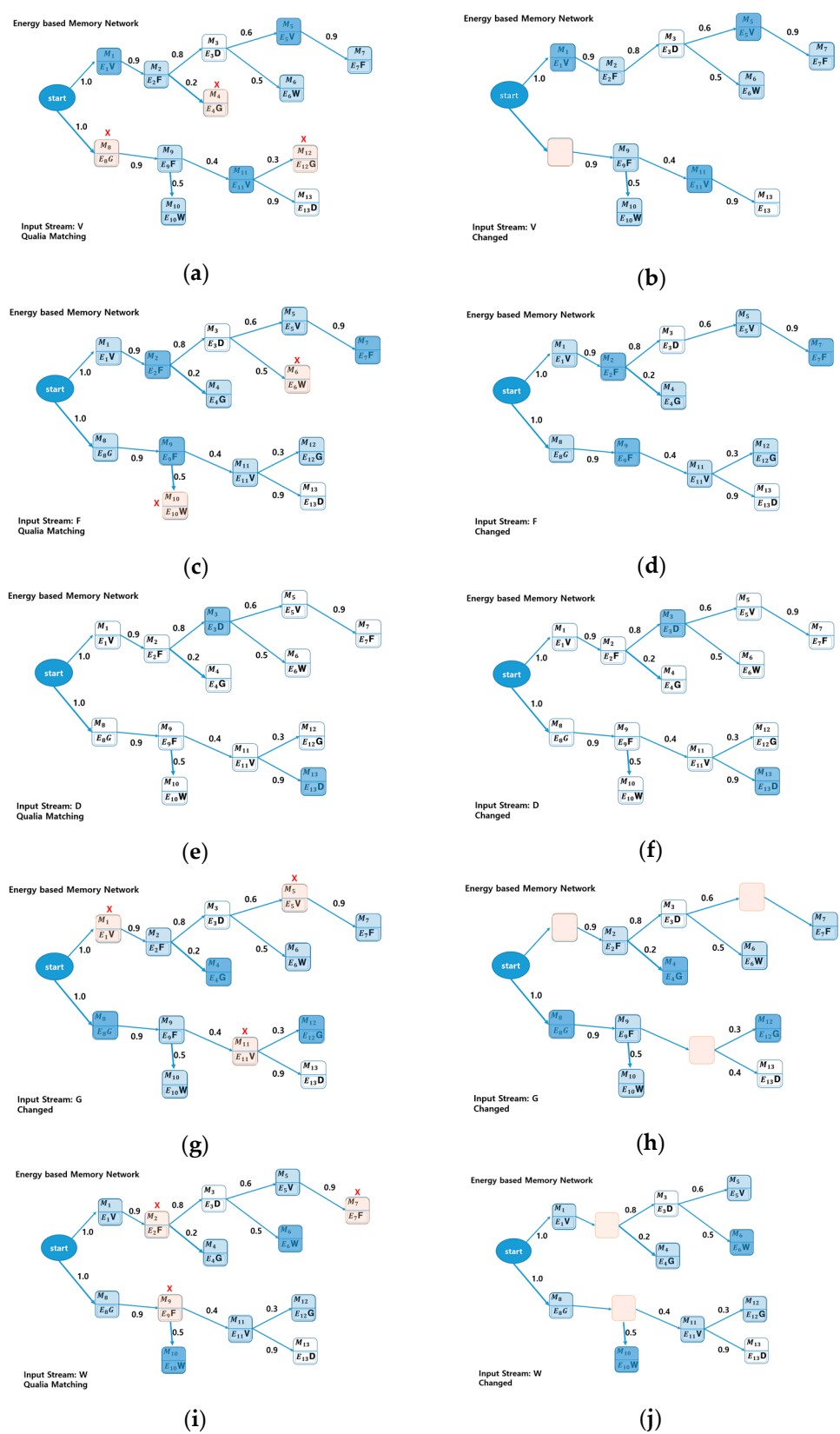

**Figure 13.** Qualia Matching and Activated Energy-based Memory network (**a**) Qualia Matching with ERS signal V (**b**) The changed state after ERS signal V coming (**c**) Qualia Matching with ERS signal F (**d**) The changed state after ERS signal F coming (**e**) Qualia Matching with ERS signal D (**f**) The changed state after ERS signal D coming (**g**) Qualia Matching with ERS signal G (**h**) The changed state after ERS signal G coming (**i**) Qualia Matching with ERS signal W (**j**) The changed state after ERS signal W coming.

The extracted Thinking Threads with qualia matching with ERS signal V in Figure 13b are as follows:

T1:　M1 0.9 M2 0.6 M5 0.9 M7 0.0 NIL
T2:　M1 0.9 M2 0.5 M6 0.0 NIL

If deep extraction is applied here, the results of depth 0.6 are as follows:

T1:　M1 0.9 M2 0.6 M5 0.0 NIL
T2:　M1 0.9 M2 0.0 NIL

Figure 14 shows the simulation results of the Thinking Thread extraction, qualia matching and deep extraction process in the case of ERS signal V. In the output list of 'M1 1.22 0.90 M2 1.08 0.60 M5 0.95 0.90 M7 1.20 0.00 NIL', 1.22 is the energy value of the memory capsule M1 and 0.90 is the value of connection link between M1 and M2. The energy values of the memory capsule are changed to adjusted values temporarily by qualia matching and incoming data energy. As shown in Figure 15, the energy values fluctuate depending on the input stream. The higher the qualia matching degree and the incoming energy value, the higher the activated degree of the memory capsule.

```
================ RESTART: D:/PAPER/MDPI/electronics/shim.py ===
* Thinking Thread Extraction from Energy based Memory Capsule ...
Qualia Matching type : V
 Thnking Thread Extraction Qualia:V
...
Thnking Thread 1
M1 1.22 0.90 M2 1.08 0.60 M5 0.95 0.90 M7 1.20 0.00 NILL
Thnking Thread 2
M1 1.22 0.90 M2 1.08 0.50 M6 0.86 0.00 NILL
... Deep Extraction
Extraction Depth?0.3
Extraction Depth : 0.3
Thnking Thread 1
M1 1.22 0.90 M2 1.08 0.60 M5 0.95 0.90 M7 1.20 0.00 NILL
Thnking Thread 2
M1 1.22 0.90 M2 1.08 0.50 M6 0.86 0.00 NILL
Extraction Depth?0.6
Extraction Depth : 0.6
Thnking Thread 1
M1 1.22 0.90 M2 1.08 0.60 M5 0.95 0.90 M7 1.20 0.00 NILL
Thnking Thread 2
M1 1.22 0.90 M2 1.08 0.00 NILL
>>> |
```

**Figure 14.** Thinking Thread Extraction by Qualia Matching and Deep Extraction.

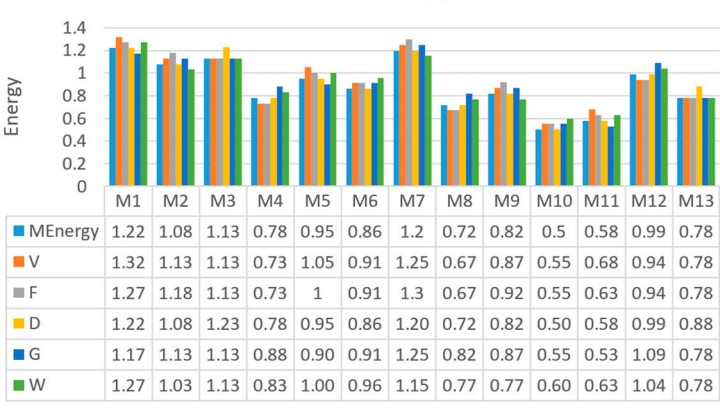

The Change of Energy Value

| | M1 | M2 | M3 | M4 | M5 | M6 | M7 | M8 | M9 | M10 | M11 | M12 | M13 |
|---|---|---|---|---|---|---|---|---|---|---|---|---|---|
| ■ MEnergy | 1.22 | 1.08 | 1.13 | 0.78 | 0.95 | 0.86 | 1.2 | 0.72 | 0.82 | 0.5 | 0.58 | 0.99 | 0.78 |
| ■ V | 1.32 | 1.13 | 1.13 | 0.73 | 1.05 | 0.91 | 1.25 | 0.67 | 0.87 | 0.55 | 0.68 | 0.94 | 0.78 |
| ■ F | 1.27 | 1.18 | 1.13 | 0.73 | 1 | 0.91 | 1.3 | 0.67 | 0.92 | 0.55 | 0.63 | 0.94 | 0.78 |
| ■ D | 1.22 | 1.08 | 1.23 | 0.78 | 0.95 | 0.86 | 1.20 | 0.72 | 0.82 | 0.50 | 0.58 | 0.99 | 0.88 |
| ■ G | 1.17 | 1.13 | 1.13 | 0.88 | 0.90 | 0.91 | 1.25 | 0.82 | 0.87 | 0.55 | 0.53 | 1.09 | 0.78 |
| ■ W | 1.27 | 1.03 | 1.13 | 0.83 | 1.00 | 0.96 | 1.15 | 0.77 | 0.77 | 0.60 | 0.63 | 1.04 | 0.78 |

**Figure 15.** The change of Energy value in Memory Capsule by Qualia Matching with V, F, D, G and W.

## 5. Conclusions

In this work, we proposed an energy-based memory network with an event-related synchronized data extraction mechanism to create a more efficient cloud database in a cloud service. We adopted the concept of qualia and firstly designed a qualia matching rule and a mechanism for feature matching. This approach means that the system can deal

with data sophisticatedly; not only the properties of things but also individual personalities or preferences. In addition, deep extraction functions were made for Thinking Thread extraction from the energy-based memory network in order to provide the extraction by level depth. In experiments, the keywords of user's demands and their qualia terms were obtained by analyzing the incoming data and the related knowledge was extracted from the energy-based memory network successfully. As a result of our test, it was confirmed that the proposed system works well. It is expected that the proposed mechanism can be a part of the core engine when constructing an intelligent system.

**Funding:** This research received no external funding.

**Conflicts of Interest:** The authors declare no conflict of interest.

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
