# Peer review of "Feature Matching Synchronized Reasoning from Energy-Based Memory Network for Intelligent Data Management in Cloud Computing Data Center"

_electronics, doi:10.3390/electronics10161900_

Round 1
Reviewer 1 Report
Thank you for the opportunity given in reviewing this interesting and current article. In my opinion, some suggestions are welcome to improve the quality of information presentation to readers as follows:
Abstract. The author does not clearly state what the main purpose or objective of this article is so that readers can quickly identify! Also, the author does not briefly mention the results and conclusions of his study!
Introduction. The same aspect mentioned above is completely missing in the introduction!
Specialty literature. I think that "related works" should be replaced with "Literature review"! I found that the literature is a bit poor in quoted sources! I suggest the author to enrich the literature review by adding other studies of specialists in the field!
We could not clearly identify where the section dedicated to research methodology and methods used to conduct research begins and ends! Maybe the author will make a separate section and try to enlighten the readers for these aspects!
The "Experiments" part is a bit abstract and very technical, maybe they should use a little more flexible language so that readers can understand the importance of experimentation!
The conclusions are a bit short and a little evasive. I suggest the author to highlight more the importance of his experiment and the implications it has on the researched field!
Author Response
Thank you very much for your good comments and pointing out the important things. I revised my paper according to your comments as follows:
- I updated the part of Abstract , Introduction and Conclusion part by adding the contents of main purpose, more detailed description of the proposed system according to your comments.
- I changed the title of ‘Related works’ to ‘Literature Review’ as you mentioned and also complemented the contents in the other section.
Please refer to the attached revised paper. I appreciate your good advice.
JeongYon Shim

Reviewer 2 Report
The paper describes a Cloud database for software as a service, with structured storage, aiming uninterrupted service. The model is inspired by human brain functions. The results presented are simulations of data retrieving operations.
The paper is well structured and the idea has some merit. Here are suggestions for improving the paper:
- Previous work and state of the art should be described in much more detail. The paper is very incomplete in that aspect. There are numerous types of memories and data structures, used for very different purposes. Properties of other memories and data structures could be discussed and compared to the model proposed, both in theory and to compare the experimental results.
- The model proposed was tested using just a user interface. That seems very far from what could serve as a cloud service. Is there anything in the model that makes it adequate and specific to saas?
- The paper focus on retrieving data from the memory. How does it work for storing data? Even if that is not explained in detail, it could at least be adressed briefly.
- Is it guaranteed that the data are retrieved if they exist?
- It should be noted that there are many problems with imitating the brain; the brain is actually unreliable and subject to many problems such as forgetting, false memories and so on. Those behaviors do not seem to be copied to the model proposed, but that could be discussed in the paper.
- Properties V, F, D, G, W of qualia could be explained. Do they have any meaningful interpretation? What is the origin of the data in table Figure 7?
- The results presented are not compared to anything. And they do not prove if the model is 100 % accurate or what is its efficiency. That is important for assessing the quality of the model.
- Figure 4 is more or less redundant, since it represents just a small part of Figure 5 and both are in fact simple diagrams.
- Acronyms should be checked. There are acronyms used before they are defined, namely SaaS. EEG, first used at line 100, is not defined.
- Very poor English, with numerous typos, extra and missing spaces, and awkward sentences. Letter capitalization must also be checked for inconsistencies (e.g. section 2.2)
- At line 44, it is a bit odd to call the human brain a product, even though a smart and efficient one.
- The references also need to be checked. There are many incomplete references.
Author Response
Thank you very much for your good comments and pointing out the important things. I revised my paper according to your comments as follows:
- I revised the contents of the part of your pointing out by adding the description about contents of previous works in more detail.
- Because this approach of the proposed system ( especially Energy based memory capsule, Qualia matching rule and mechanism, deep extraction by level depth and etc.) is a new one, it is not easy to compare with other systems. I couldn’t find a similar system which can be easily compared to this proposed method yet, so I focused on description about system mechanism. Please understand this point.
- The mechanism for implementing the intelligent Cloud Data Base as SaaS is designed.
- This paper is focusing on the data retrieval. About memory storage and management, I add the brief description in the section 3.2.4 and introduced to refer my other paper.
- The redundant Figure 4 was removed.
- I complemented the description about the concept of Qualia and its terms. Also Experiments , conclusion and Reference part were revised.
Please refer to the attached revised paper. I appreciate your good advice.
JeongYon Shim

Reviewer 3 Report
- Related works section is not enough such as they talked about cloud computing and simple synchronization but I have not found any related literature on feature matching synchronization. Add some recent year papers in related work, mostly references are not recent ones.
- Manuscript has a lot of grammatical mistakes, such as on lines 84-87, 346-360.
- Somewhere they used term Quailia, sometimes they used term Qualia, difficult to understand the context, I wonder if these terms are same meanings if same then correct it
- Section 3.3.3 has a lot of sentence construction mistakes, need to be corrected,
- In algorithm 1, authors just wrote that, as it is, there is no description of algorithm in text section, it should be described well, it is very difficult to understand that, also in the algorithm mathematical styling should be used while writing algorithms variables.
- Discussion on experiments is not enough and rich, they just wrote that this table contains these values, this figure contains this values etc, they should have discuss the impacts and variation in values, and why that variation values etc.
- Conclusion section needs overhaul, there is on comparison of proposed approach with any baseline or prior study approaches, if it is first of its kind, then authors should need to write and clarify in conclusion section
- Conclusion lines specifically 478-480, a lot of spelling mistakes and sentence construction mistakes.
- Overall, manuscript needs improvements, sometime spelling mistakes, sentence construction mistakes, break the tempo to review the manuscript.
Author Response
Thank you very much for your good comments and pointing out the important things. I revised my paper according to your comments as follows:
- I revised the description about ‘Qualia’ and its terms in more detail.
- The algorithm describes the procedure of program and the related mathematical formulas are specified and explained in the body contents of section.
- The spelling and grammar check were made and updated.
Please refer to the attached revised paper. I appreciate your good advice.
JeongYon Shim

Round 2
Reviewer 1 Report
Something is wrong with the distribution of bibliographic references! At the beginning, the bibliographic notes from 1 to 13 are missing !!!
Please check the correct order of the bibliography references in the text of the article!
Author Response
Thank you very much for your comments. I revised the distribution of bibliographic references according to your guideline. Please check my revised paper. Thanks a lot!
Best Regards,
JeongYon Shim
Reviewer 2 Report
The paper still has some weaknesses but it is in general better than before.
The correct multiplication sign should be used, not asterisk.
Letter capitalization should be revised in the algorithms (and the algorithms better formatted).
Author Response
Thank you very much for your good comments. I updated multiplication sign and revised the algorithm format. Please check my revised paper. Thanks a lot!
Best Regards,
JeongYon Shim
Reviewer 3 Report
The rebuttal was satisfactory. I have no further questions. Thanks
Author Response
Thank you very much for your review and good comments.
Best Regards,
JeongYon Shim